# Autoantibodies to PAX5, PTCH1, and GNA11 as Serological Biomarkers in the Detection of Hepatocellular Carcinoma in Hispanic Americans

**DOI:** 10.3390/ijms24043721

**Published:** 2023-02-13

**Authors:** Cuipeng Qiu, Yangcheng Ma, Bofei Wang, Xiaojun Zhang, Xiao Wang, Jian-Ying Zhang

**Affiliations:** 1Department of Biological Sciences & NIH-Sponsored Border Biomedical Research Center, The University of Texas at El Paso, El Paso, TX 79968, USA; 2Department of Leukemia, The University of Texas at MD Anderson Cancer Center, Houston, TX 77030, USA

**Keywords:** autoantibody, tumor-associated antigen, hepatocellular carcinoma, biomarker, Hispanic Americans

## Abstract

Studies have demonstrated that autoantibodies to tumor-associated antigens (TAAs) may be used as efficient biomarkers with low-cost and highly sensitive characteristics. In this study, an enzyme-linked immunosorbent assay (ELISA) was conducted to analyze autoantibodies to paired box protein Pax-5 (PAX5), protein patched homolog 1 (PTCH1), and guanine nucleotide-binding protein subunit alpha-11 (GNA11) in sera from Hispanic Americans including hepatocellular carcinoma (HCC) patients, patients with liver cirrhosis (LC), patients with chronic hepatitis (CH), as well as normal controls. Meanwhile, 33 serial sera from eight HCC patients before and after diagnosis were used to explore the potential of these three autoantibodies as early biomarkers. In addition, an independent non-Hispanic cohort was used to evaluate the specificity of these three autoantibodies. In the Hispanic cohort, at the 95.0% specificity for healthy controls, 52.0%, 44.0%, and 44.0% of HCC patients showed significantly elevated levels of autoantibodies to PAX5, PTCH1, and GNA11, respectively. Among patients with LC, the frequencies for autoantibodies to PAX5, PTCH1, and GNA11 were 32.1%, 35.7%, and 25.0%, respectively. The area under the ROC curves (AUCs) of autoantibodies to PAX5, PTCH1, and GNA11 for identifying HCC from healthy controls were 0.908, 0.924, and 0.913, respectively. When these three autoantibodies were combined as a panel, the sensitivity could be improved to 68%. The prevalence of PAX5, PTCH1, and GNA11 autoantibodies has already occurred in 62.5%, 62.5%, or 75.0% of patients before clinical diagnosis, respectively. In the non-Hispanic cohort, autoantibodies to PTCH1 showed no significant difference; however, autoantibodies to PAX5, PTCH1, and GNA11 showed potential value as biomarkers for early detection of HCC in the Hispanic population and they may monitor the transition of patients with high-risk (LC, CH) to HCC. Using a panel of the three anti-TAA autoantibodies may enhance the detection of HCC.

## 1. Introduction

Hepatocellular carcinoma (HCC) remains the third leading cause of cancer death and the sixth most commonly diagnosed cancer worldwide in 2020 [1]. In the United States, it is reported that HCC in Hispanic Americans will continue to increase [2]. The incidence rates in Hispanics are two times higher compared to White non-Hispanics [3]. Serological markers for HCC testing, the alpha-fetoprotein (AFP) and des-gamma-carboxy prothrombin (DGCP), also known as Prothrombin Induced by Vitamin K Absence II (PIVKA II), have been well-studied. However, neither AFP nor DGCP are adequate serological markers for HCC screening tests [4]. Therefore, exploring other biomarkers that may improve sensitivity and specificity as a supplementary tool for HCC detection are needed. Autoantibodies triggered by tumor-associated antigens (TAAs) are promising biomarkers that are stable and easily detectable in the blood. The immune amplification effect leads to an abundance of these autoantibodies appearing, which could facilitate the use of autoantibodies in the detection of cancer. The development of cancer involves the accumulation of gene mutations that affect the expression of proteins. Those abnormally expressed proteins may provoke the immune response to induce the corresponding autoantibodies. For this reason, it is important to identify the tumor-associated proteins and to further evaluate the autoantibodies against these proteins.

Paired box protein Pax-5, also known as B-cell specific activator protein (BSAP), is a transcription factor involved in the control of organ development and tissue differentiation [5]. It plays an important role in the hematopoietic system, not only on differentiation and maturation of B-cells, but also the inhibition of differentiation and progress toward other lineages [6]. It is reported that the loss of *PAX5* leads to B-cell malignancies in mouse models, and the mutation of *PAX5* in humans causes genetic lesions in acute lymphoblastic leukemia [7,8]. Studies have shown that PAX5 is expressed in Hodgkin and B-cell non-Hodgkin lymphomas, precursor B-cell lymphoblastic neoplasms, medulloblastoma, Merkel cell carcinoma, differentiated neuroendocrine carcinoma of the digestive system, small cell lung carcinoma, and rhabdomyosarcoma, suggesting that it might be a biomarker for the diagnosis or prognosis of those cancers [5,9,10,11,12,13]. Moreover, PAX5 is a functional tumor suppressor that is involved in HCC carcinogenesis by regulation of the p53 signaling pathway [14,15]. Patched 1 (PTCH1) is a transmembrane receptor protein that suppresses the hedgehog signaling, which is involved in the tumorigenesis of human tumors [16,17]. The PTCH1 protein is highly expressed in bladder cancers compared to adjacent normal bladder tissues and has potential value as a prognostic biomarker of bladder cancer [18]. Elevated PTCH1 at protein and mRNA levels might have the potential to determine colorectal cancer with high-risk metastasis from that with low-risk metastasis [19]. In addition, PTCH1 gene mutation or autoantibodies to PTCH1 could be predictors of breast cancer [20,21]. GNA11, the subunit of G proteins, is involved as modulators or transducers in various transmembrane signaling systems [22]. The gene mutations of GNA11 were frequently observed in human uveal melanomas, implying that it could lead to the initiation of uveal melanomas [23,24,25]. A study showed that GNA11 mutations also occurred in hepatic small vessel neoplasms [26]. Autoantibodies to GNA11 were also found in the sera of patients with esophageal squamous cell carcinoma and patients with lung cancer, and may serve as biomarkers for early detection in those two types of cancers [27,28].

Proteomic profiling could aid in the development of more accurate biomarkers for HCC detection [29,30]. Autoantibodies against tumor-associated antigens (TAAs) are promising biomarkers that have been supported by previous research [31,32,33,34,35]. Although biomarkers related to cancer are well studied, autoantibodies to PAX5, PTCH1, and GNA11 were reported in a previous study where their existence and specificity among different ethnic populations were underreported. Moreover, their detection in patients who are transitioning from high-risk to HCC has not been seen yet. Based on a recent study that evaluated multiple autoantibodies in HCC detection [36], we reported three autoantibodies to PAX5, GNA11, and PTCH1 with improved performances and explored their potential in the detection of HCC in the Hispanic American population and non-Hispanic population. We further explored the prevalence of those three anti-TAA autoantibodies in a serial sera of patients before and after diagnosis with HCC to confirm their potential as early detection markers.

## 2. Results

### 2.1. Expression of Autoantibodies to PAX5, PTCH1, and GNA11 in Different Groups of Subjects

The basic information of PAX5, PTCH1, and GNA11 is shown in Table 1. They are all involved in core pathways related to cancer development. To measure the appearance of autoantibodies to three TAAs in sera from HCC, LC, and CH patients, and normal controls, the indirect ELISA was performed. As shown in Figure 1, the three autoantibodies show the same trend, in which their expression levels gradually rise from CH and LC to HCC groups and they are all significantly higher (*p* < 0.0001) than the NC group as well as the CH group. The highest level of all three autoantibodies is observed in the HCC group (Figure 1).

After we set the constraints on the positive number of control subjects to less than 5%, as shown in Figure 2A, there are 52.0% of HCC sera with positive autoantibody reaction to PAX5, which is significantly higher than those of the CH sera (22.2%) and NC (4.7%) groups. Additionally, autoantibodies to PTCH1 and GNA11 are detected in 44.0% HCC sera, which are significantly higher than those in CH sera (18.5% to GNA11, 14.8% to GNA11) and NC sera (4.7%). The frequencies of autoantibodies to PAX5, PTCH1, and GNA11 in the LC group are 32.1%, 35.7%, and 25.0%, which are significantly higher than their frequencies in the CH and NC groups (Figure 2B,C). There is no statistical significance observed in frequencies of any of three autoantibodies between the HCC and LC groups, but when a parallel test is performed to combine the three autoantibodies, 68.0% of the HCC patients are identified, which is significantly higher than the LC (39.3%), CH (22.2), and NC (5.9%) groups (Figure 2D). Additionally, the frequency of the combination of those three autoantibodies in the LC group is higher than that in the NC group; there is no significant difference observed in the frequency between the CH and NC groups (Figure 2D).

### 2.2. Performance of Autoantibodies to PAX5, PTCH1, and GNA11 in the Detection of HCC

The autoantibodies to PAX5, PTCH1, and GNA11 have high classification abilities between the HCC group and NC group (Figure 3A–C). The autoantibody to PAX5 yields an area under the ROC curve (AUC) of 0.908, with a sensitivity of 52.0% at a 95.3% specificity to identify HCC from NC. In addition, the autoantibodies to PTCH1 and GNA11 show AUCs of 0.913 (44.0% of sensitivity) and 0.924 (56.0% of sensitivity) with a 95.3% specificity to discriminate between HCC and NC, respectively. The autoantibody to PTCH1 has the highest AUC value to identify HCC from NC compared with the others (Figure 3B). To further evaluate the performance of these three autoantibodies, the parameters such as the false positive rate (FPR), false negative rate (FNR), positive predictive value (PPV), negative predictive value (NPV), positive likelihood-ratio (+LR), negative likelihood ratio (-LR), Youden index (YI), and accuracy were applied. As shown in Table 2, it is observed that, with the addition of the three autoantibodies as a parallel test, sensitivity, PPV, NPV, YI, and accuracy are all increased with a slightly decreased specificity. The combination of those three autoantibodies shows a sensitivity of 68.0%, a specificity of 94.1%, and an accuracy of 88.2%, which are all higher than those in an individual autoantibody. The combination also has a relatively lower FNR, which indicates that it could identify more HCC patients (Table 2).

In addition, the three autoantibodies may have the potential to identify HCC from CH populations. As shown in Figure 3D–F, the autoantibody to GNA11 has the highest AUC of 0.768 to distinguish between HCC and CH, with 20.0% sensitivity and 96.3% specificity. The autoantibody to PAX5 could identify 12.1% HCC from CH with 96.3% specificity and an AUC of 0.741. The autoantibody to PTCH1 yields an AUC of 0.760 with 8.0% sensitivity and 96.3% specificity to identify HCC from LC.

### 2.3. Autoantibodies to PAX5, PTCH1, and GNA11 Could Identify LC from NC

Since patients from the LC group are at high risk of turning into HCC, it is necessary to explore the potential of these three anti-TAA autoantibodies to distinguish patients at high risk from healthy subjects. The results show that all three autoantibodies have moderate classification abilities to identify patients at high risk of HCC from normal controls. As indicated in Figure 3G–I, for the identification of LC from NC, the AUCs of autoantibodies to PAX5, PTCH1, and GNA11 are 0.825, 0.861, and 0.840, respectively. At the 95.3% specificity of NC, the sensitivities are 35.7%, 39.3%, and 25.0%, respectively.

### 2.4. Autoantibodies to PAX5, PTCH1, and GNA11 Elevated in HCC Patients before Clinical Diagnosis

To further explore whether the three autoantibodies could be early HCC biomarkers, we tested them by using 33 serial sera from eight HCC patients who have been followed up before and after diagnosis of HCC for more than one year. Three to six sera from each of the eight HCC patients were collected every three months before and after diagnosis. The dynamic changes of levels of those three autoantibodies are shown in Figure 4, five of eight cases (62.5%) cases show the positive autoantibody reaction to PAX5 and PTCH1 before diagnosis with HCC. Among these five positive patients, two (Case1, Case6) or three (Case1, Case4, and Case6) show negative autoantibody reactions to PAX5 and PTCH1 after diagnosis with HCC, respectively. Additionally, the autoantibody to GNA11 appears in six of eight (75.0%) cases before diagnosis, while it drops to normal in Case1, Case6, and Case7 after diagnosis with HCC (Figure 4).

### 2.5. Specificity Test of Autoantibodies to PAX5, PTCH1, and GNA11

An independent non-Hispanic cohort was also used to evaluate whether these three autoantibodies were expressed differently in two ethnic cohorts. As shown in Figure 5, the autoantibodies to PAX5 and GNA11 show significantly higher frequencies in the HCC (18.9%, 32.1%, respectively) and LC (28.6% for both) groups than in the NC (4.7%) group. However, the autoantibody to PTCH1 has no significant difference between the HCC patients and normal controls in the non-Hispanic cohort, but it has a significant difference between HCC patients and normal controls in the Hispanic cohort. Therefore, the autoantibody to PTCH1 might be more specific to the Hispanic cohort.

## 3. Discussion

Cirrhosis and hepatitis B virus (HBV) infection are common risk factors of HCC [37]. In the United States, HCC remains the fastest growing mortality rate of cancer, especially along the Mexican border in Texas [38]. It is forecasted that by 2030, Hispanic men and Black women will have the highest incident rates of HCC in the United States [39]. Due to the high degree of tumor heterogeneity, HCC exhibits a poor prognostic outcome, and the five-year survival rate of HCC is less than 15% [40,41]. Therefore, early detection and screening of HCC in high-risk populations are an impactful approach to improve HCC prognosis [42]. As one of the blood-based biomarkers, anti-TAA autoantibodies hold the merits of stability and appearing earlier than a clinical diagnosis; hence, they show great potential for the early detection of HCC [43]. In this study, we explored the diagnostic potential of three autoantibodies (PAX5, PTCH1, GNA11) as serological biomarkers for the early detection of HCC patients in Hispanic Americans. Our results showed the existence of these three autoantibodies in the sera of patients with HCC, LC, and CH; HCC patients had the highest expression level, followed by the LC and CH group, which means those autoantibodies appeared not only in HCC patients, but also in patients with LC or CH who may develop to HCC. Our findings suggest that the autoantibodies to PAX5, PTCH1, and GNA11 might be indicators for early malignancy of HCC.

A study reported that anti-PAX5, anti-PTCH1, and anti-GNA11 autoantibodies have sensitivities of 22.5%, 13.3%, and 31.7% to detect HCC [36]. Another research showed that the AUC of an anti-GNA11 autoantibody for distinguishing lung cancer from normal controls was 0.724 [28]. In the present study, based on the evaluation of autoantibodies to PAX5, PTCH1, and GNA11 in Hispanic American subjects, the three autoantibodies have high diagnostic performances to identify HCC from NC with AUCs of 0.908, 0.913, and 0.924, respectively. They also have moderate classification abilities to identify HCC from CH with AUCs of 0.740, 0.760, and 0.768, respectively. Although those three autoantibodies may not differentiate HCC from LC, they still have relatively higher levels in HCC compared with LC. Moreover, all three autoantibodies have the highest frequencies in HCC (56.0%, 44.0%, 44.0%) compared with those in LC (32.1%, 35.7%, 25.7%) and CH (22.2%, 18.5%, 14.8%). In addition, these three autoantibodies may also identify LC from NC with moderate performance. Therefore, the autoantibodies to PAX5, PTCH1, and GNA11 have great potential for the early detection or differential diagnosis of HCC in Hispanic Americans. Further work with a larger sample size is required to confirm the performance of those three autoantibodies for the early detection of Hispanic HCC.

Studies have suggested that the combination of multiple anti-TAA autoantibodies as a panel could enhance the detection of cancer [20,44,45]. This study combined three anti-TAA autoantibodies as a panel; the panel could identify 68.0% HCC from NC with an accuracy of 88.2%, which was higher than those in any single one of the three autoantibodies. The frequency of the panel in the HCC group (68.0%) was significantly higher than that in the LC (39.3%) or CH (22.2%) groups. Thus, a panel of the three anti-TAA autoantibodies has better performance for the detection of HCC, and to some extent it could also monitor precancerous lesions such as LC and CH, which may progress to HCC. By evaluating the changes of these three autoantibodies in the clinical follow-up sera of HCC patients before and after diagnosis, it was found that the levels of these three autoantibodies were already elevated in sera collected nine, six, and three months before being diagnosed with HCC. Our findings are supported by studies on biomarkers for HCC and gastric cancer, which also revealed that autoantibodies existed in pre-malignant lesions [31,46,47].

To further explore the specificity of the three autoantibodies to Hispanics, we tested them in a non-Hispanic cohort. We found that the levels of the autoantibody to PTCH1 only showed a significant difference in the Hispanic cohort but not in the non-Hispanic cohort, which means it could be more sensitive to the Hispanic population. The autoantibodies to PAX5 and GNA11 were also significantly expressed in the non-Hispanic cohort. Thus, we can obtain a clue that the autoantibody to PTCH1 might be specific to the Hispanic population, whereas the mechanism of this result still needs more research. We speculate that this may be related to the mutations of the PTCH1 gene in Hispanic populations. However, this study has some limitations. First, since this is a preliminary study, the Hispanic sample size was not large enough for its clinical use. Second, the sera from HCC patients before and after diagnosis were followed up for one to two years, and a longer period of serial serum samples would be more solid to reflect the dynamic changes of the three autoantibodies. Further validations with a larger sample size are needed to address the potential of these three autoantibodies for clinical application.

## 4. Materials and Methods

### 4.1. Study Subjects

A total of 364 serum samples were used in this study. Hispanic sera from 25 patients with HCC, 28 patients with liver cirrhosis (LC), and 27 patients with chronic hepatitis (CH), as well as 85 normal human sera as normal controls (NC) were obtained from the serum bank of Cancer Autoimmunity and Epidemiology Research Laboratory at UTEP (University of Texas at El Paso), Texas. Normal controls had no obvious evidence of malignancy. From the same serum bank, a non-Hispanic cohort including 53 HCC, 35 LC, 35 CH, and 43 NC were used for the specificity test, and 33 serial serum samples from 8 HCC patients were also used to explore the potential of three autoantibodies in the early detection of HCC. For each patient, at least two to six serial samples were collected several months prior to clinical diagnosis of liver malignancy. All of the patients had a previous history of chronic hepatitis or liver cirrhosis. Due to regulations concerning studies of human subjects, the patients’ names and identification numbers were blinded to the investigators and some clinical information for sera used in this study was unavailable. This study was approved by the Institutional Review Board of the University of Texas at El Paso.

### 4.2. Recombinant Proteins and Enzyme-Linked Immunosorbent Assay (ELISA)

Three recombinant proteins (PAX5, PTCH1, GNA11) were commercially purchased from the LD Biopharma Inc (San Diego, CA, USA). The purity of the three recombinant proteins was confirmed by SDS-PAGE gel. The expression of autoantibodies to PAX5, PTCH1, and GNA11 was tested by indirect ELISA. Purified proteins were diluted in phosphate-buffered saline (PBS) with an optimal concentration (0.5 μg/mL for PTCH1 and GNA11, 0.25 μg/mL for PAX5) for coating 96-well Immunolon microtiter plates (Fisher Scientific, Huston, TX, USA) overnight at 4 °C. The coated plates were incubated with a post-coating solution for 2 h at room temperature. After washing with PBS containing 0.05% Tween-20 (PBST), the coated plates were incubated with the serum samples diluted at 1:200 for 1 h in 37 °C water bath. Coated plates were washed with PBST, then the plates were incubated with the horseradish peroxidase (HRP)-conjugated goat anti-human IgG (Santa Cruz Biotechnology, Inc. Santa Cruz, CA, USA) diluted at 1:3000 for 1 h in 37 °C water bath followed by washing with PBST. The substrate 2,2′-azino-bis (3-ethyl-benzothiazoline-6-sulfonic acid) diammonium salt (ABTS) (Sigma, St. Louis, MO, USA) was used as the detecting agent. The optical density (OD) value of each well was read at 405 nm. In addition, eight sera (1:200 dilution) were added to each plate for the normalization among different plates. If the coefficient of the variation (CV) value among different plates was more than 15%, the experiment was reran.

### 4.3. Statistical Analysis

The Mann–Whitney U test and Kruskal–Wallis H test were used to compare the difference in OD values among different groups. The Chi-square test and Fisher’s exact test were performed to compare the frequency of autoantibodies in different groups. The receiver operating characteristics (ROC) curve analysis was used to evaluate the performance of three autoantibodies in the detection of HCC as well as to determine the cut-off values. In addition, performance indexes including the sensitivity, specificity, false positive rate (FPR), false negative rate (FNR), Youden index (YI), positive likelihood-ratio (+LR), negative likelihood ratio (−LR), accuracy, positive predictive value (PPV) and negative predictive value (NPV) was calculated to evaluate the validity and reliability of these three autoantibodies as potential biomarkers. The cut-off value was determined according to FPR < 5% (specificity > 95%).

## 5. Conclusions

In summary, this is the first study to evaluate autoantibodies to PAX5, PTCH1, and GNA11 in the Hispanic population including different groups of patients corresponding to various HCC development stages. Our results indicate that the three anti-TAAs autoantibodies hold great diagnostic value for the early detection of HCC in Hispanic Americans. Furthermore, the autoantibody to PTCH1 might be specific to Hispanic populations. These three autoantibodies have already appeared in pre-malignant lesions prior to clinical diagnosis. Given their presentation in patients with pre-malignant lesions, they might monitor the transition of patients with high-risk (LC and CH) to HCC. Additionally, the combination of the three autoantibodies showed higher performance to detect HCC, suggesting that the panel might contribute to the identification of HCC in high-risk individuals by enhancing the detection accuracy. Future studies may focus on the performance of the panel in a large cohort study to further confirm the value in high-risk populations.

## Figures and Tables

**Figure 1 ijms-24-03721-f001:**
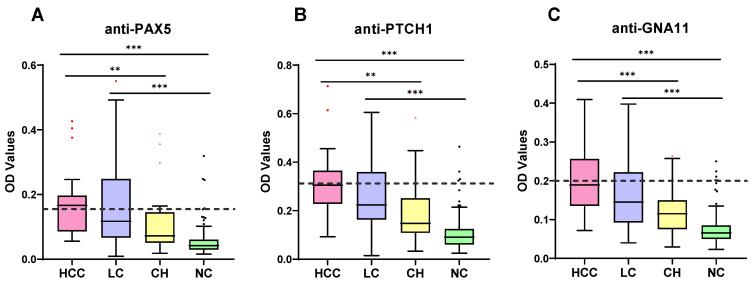
Expression of autoantibodies to PAX5 (**A**), PTCH1 (**B**), and GNA11 (**C**)in different subjects including HCC patients, high-risk patients (LC and CH), and normal controls. HCC, hepatocellular carcinoma; LC, liver cirrhosis; CH, chronic hepatitis; NC, normal controls. Cut-off values of individual autoantibodies are indicated by dotted lines. **, *p* < 0.01; ***, *p* < 0.001.

**Figure 2 ijms-24-03721-f002:**
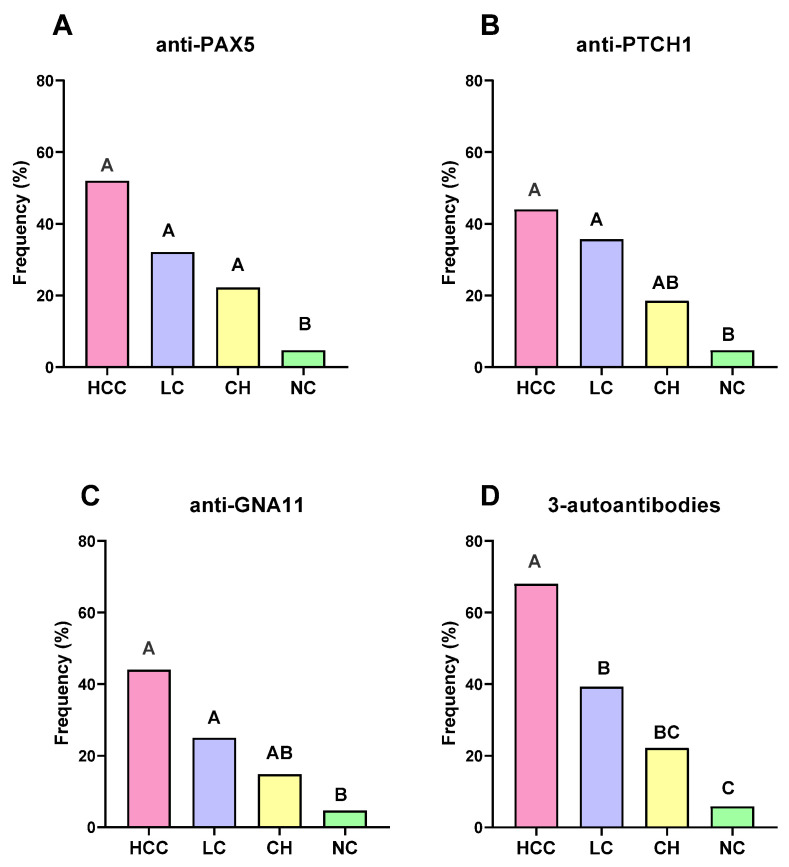
Frequencies of autoantibodies to PAX5 (**A**), PTCH1 (**B**), GNA11 (**C**) , and the combination of these three autoantibodies (**D**) in different study subjects. HCC, hepatocellular carcinoma; LC, liver cirrhosis; CH, chronic hepatitis; and NC, normal controls. The significant difference was indicated if the letters above the bars are totally different (e.g.,: A > BC, A > B, A > C, B > C).

**Figure 3 ijms-24-03721-f003:**
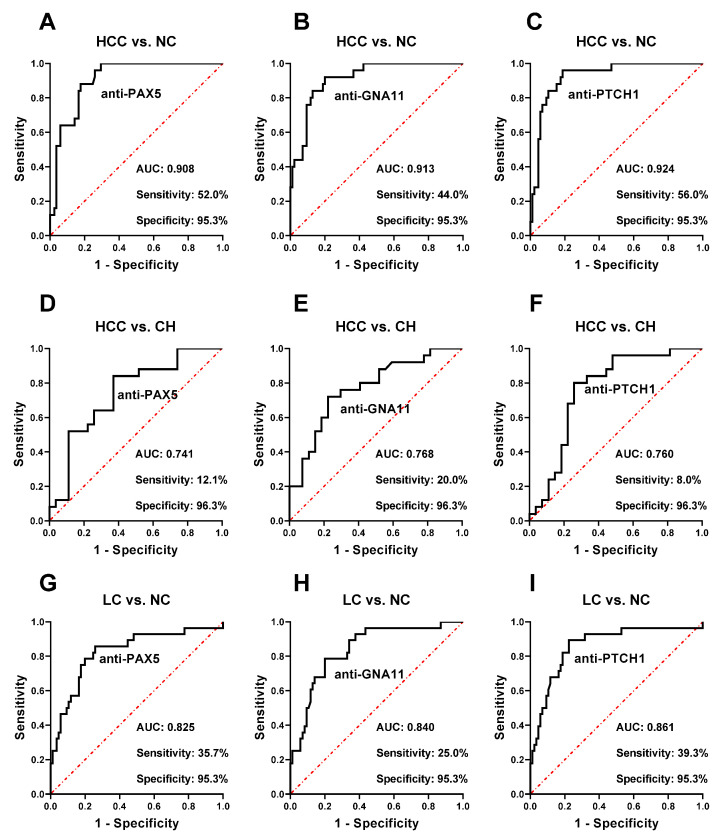
Performance of autoantibodies to PAX5, PTCH1, and GNA11 in the detection of HCC and LC. (**A**–**C**) performance of three autoantibodies in the identification of HCC from NC; (**D**–**F**) performance of three autoantibodies in the identification of HCC from CH; (**G**–**I**) performance of three autoantibodies in the identification of LC from NC. HCC, hepatocellular carcinoma; LC, liver cirrhosis; CH, chronic hepatitis; and NC, normal controls.

**Figure 4 ijms-24-03721-f004:**
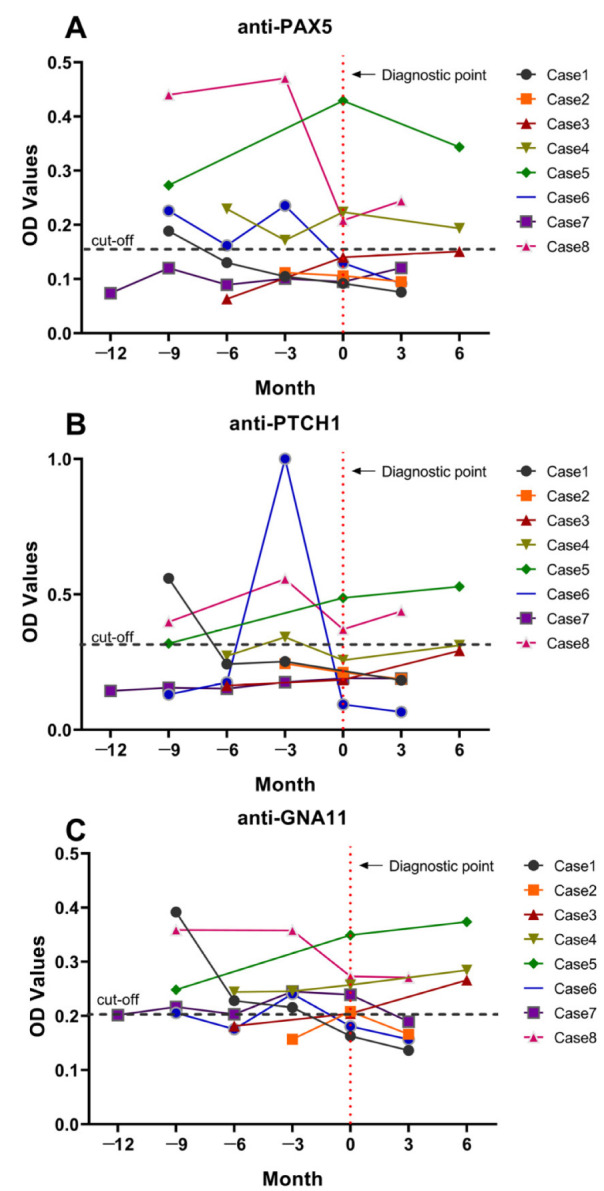
Levels of autoantibodies to PAX5 (**A**), PTCH1 (**B**), and GNA11 (**C**) in the sera of eight HCC patients who have been followed up before and after diagnosis of HCC for more than one year. Cut-off values of individual autoantibodies are shown by black dotted lines. The red dotted line indicates the time point where patients were clinically diagnosed with HCC.

**Figure 5 ijms-24-03721-f005:**
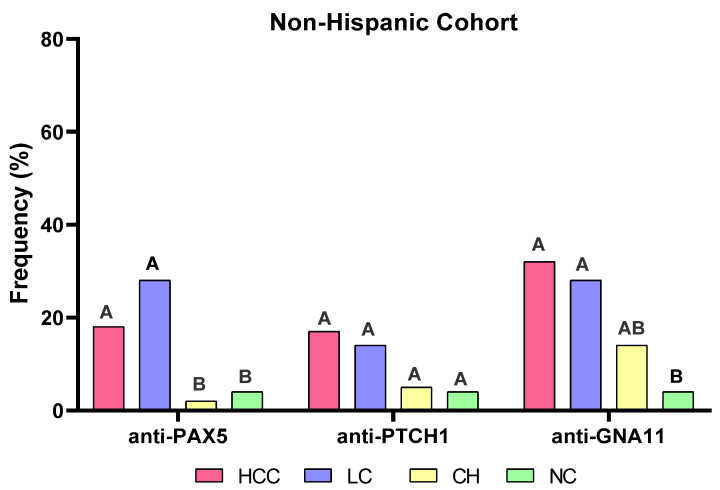
Frequencies of autoantibodies to PAX5, PTCH1, and GNA11 in an independent non-Hispanic cohort. HCC, hepatocellular carcinoma; LC, liver cirrhosis; CH, chronic hepatitis; NC, normal controls. The significant difference was indicated if the letters above the bars are totally different (e.g.,: A > B).

**Table 1 ijms-24-03721-t001:** Characteristics of PAX5, PTCH1, and GNA11.

TAAs	UniProt ID	Full Protein Name	Gene Classification	Core Pathway	Process	Function *
PAX5	Q02548	Paired box protein Pax-5	TSG	Chromatin Modification	Cell Fate	DNA-binding transcription activator activity; DNA-binding transcription factor activity
PTCH1	Q13635	Protein patched homolog 1	TSG	HH	Cell Fate	Cholesterol binding, cell differentiation, liver regeneration
GNA11	P29992	Guanine nucleotide-binding protein subunit alpha-11	Oncogene	PI3K; RAS; MAPK	Cell Survival	G-protein beta/gamma-subunit complex binding; G-protein-coupled receptor binding; GTP binding

*: The function of proteins was obtained from UniProt (https://www.uniprot.org/, accessed on 4 May 2022). TSG, Tumor suppresser gene. HH, Hedgehog.

**Table 2 ijms-24-03721-t002:** Performance of the combination of autoantibodies to PAX5, PTCH1, and GNA11.

Autoantibodies	Se (%)	Sp (%)	FPR (%)	FNR (%)	PPV (%)	NPV (%)	+LR	−LR	YI	Accuracy (%)
Anti-PAX5	52.0	95.3	4.7	48.0	76.5	87.1	11.06	0.50	0.47	85.5
Anti-PAX5 or anti-PTCH1	64.0	94.1	5.9	36.0	76.2	89.9	10.85	0.38	0.58	87.3
Anti-PAX5 or anti-PTCH1 or anti-GNA11	68.0	94.1	5.9	32.0	77.3	91.7	11.53	0.34	0.62	88.2

Se, sensitivity; Sp, specificity; FPR, false positive rate; FNR, false negative rate; PPV, positive predictive value; NPV, negative predictive value; +LR, positive likelihood-ratio; −LR, negative likelihood ratio; YI, Youden index.

## Data Availability

Data are available on request to the corresponding author.

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
