# Peer review of "Autoantibodies to PAX5, PTCH1, and GNA11 as Serological Biomarkers in the Detection of Hepatocellular Carcinoma in Hispanic Americans"

_ijms, 2023, doi:10.3390/ijms24043721_

Round 1

Reviewer 1 Report (New Reviewer)

The study was well done and interesting, but there were points to be reconsidered:

·         Is the assay of autoantibodies ( PAX5, PTCH1, and GNA11) validated?

·         What the authors mean by clinical diagnosis of liver malignancy and which laboratory tests were used for clinical diagnosis of malignancy or by another way how the authors confirmed the serum samples that were used in their studies for patients with malignancy?

·         In relation to the serum samples from 8 HCC patients that were used to explore the potential of three autoantibodies in the early detection of HCC and for r each patient, serum was collected several months prior to clinical diagnosis of liver malignancy

1-  At which stage of HCC did you collect the sample as confirmed histopathologically?

2- For the appearance of clinical signs of HCC it takes a long period of time and you, collect serum several months prior to the clinical diagnosis of liver malignancy, preservation of serum samples for long period degrades the antigens (PAX5, PTCH1, and GNA11)   and this may affect the reliability of your results. 

Author Response

Dear Editors and Reviewers:

Thank you for your comments regarding our manuscript titled “Autoantibodies to PAX5, PTCH1, and GNA11 as serological biomarkers in the detection of hepatocellular carcinoma in Hispanic Americans”. We have read your comments carefully and made significant revisions accordingly. We have also added validation data to this manuscript. Our additional validation cohort was used to confirm the performance of the three autoantibodies in the detection of hepatocellular carcinoma in Hispanic patients. In addition, more contents were added to the manuscript. Revised portions are marked in red in the manuscript and responses to the reviewers’ comments are as follows:

Response to Reviewer 1.

Comments and Suggestions for Authors

The study was well done and interesting, but there were points to be reconsidered:

  1. Is the assay of autoantibodies (PAX5, PTCH1, and GNA11) validated?

Response: We thank the reviewer for this comment. We have added an independent cohort including 53 HCC (hepatocellular carcinoma), 35 LC (liver cirrhosis), 35 CH (chronic hepatitis), and 43 NC (normal controls) to our study. Unfortunately, there are no more Hispanic sera available in our serum bank, so the additional non-Hispanic sera were used for further validation. Interestingly, one of the three autoantibodies has no significant difference among the HCC patients at different stages and normal controls in non-Hispanics, which may further indicate that this autoantibody might be specific to Hispanics.

  1. What the authors mean by clinical diagnosis of liver malignancy and which laboratory tests were used for clinical diagnosis of malignancy or by another way how the authors confirmed the serum samples that were used in their studies for patients with malignancy?

Response: We thank the reviewer for this comment. The samples were provided by our collaborator who collected these sera from the hospital. We were told by our collaborators that the diagnosis of liver malignancy is based on the standard protocol of the hospital.

  1. In relation to the serum samples from 8 HCC patients that were used to explore the potential of three autoantibodies in the early detection of HCC and for each patient, serum was collected several months prior to clinical diagnosis of liver malignancy

1). At which stage of HCC did you collect the sample as confirmed histopathologically?

Response: We thank the reviewer for this comment. Serial sera of HCC patients were provided by our collaborator. Patients with HCC or precancerous lesion were confirmed by our collaborator. However, due to regulations concerning studies of human subjects, the patient's name and identification number were blinded to the investigators and some clinical information for the sera used in this study was unavailable. We do not know the stage of HCC during which our samples were collected. It was difficult to follow up patients over a long period of time to obtain serum at each stage. In a future study with a larger cohort, we hope to confirm our findings.

2). For the appearance of clinical signs of HCC it takes a long period of time and you, collect serum several months prior to the clinical diagnosis of liver malignancy, preservation of serum samples for long period degrades the antigens (PAX5, PTCH1, and GNA11), and this may affect the reliability of your results.

Response: We thank the reviewer for this comment. We understand that it takes a long period of time for patients from precancerous lesion develop to HCC, and it is best to collect serum samples from patients regularly throughout the development of HCC. Serum samples over long time periods are the most reliable, but they are difficult to obtain. We were fortunate that there were some HCC serial sera available in our serum bank. Even though the titer of autoantibodies in serum might decrease over time due to degradation, all sera from different groups such as before and after diagnosis were subject to same experience and impact without trend change. In this study, we reported our findings based on all the serum samples that were available in our sera bank. We also included liver cirrhosis and chronic hepatitis sera as controls to further support our results highlighting the potential of three autoantibodies as early HCC biomarkers. Moreover, previous studies showed that autoantibodies increased several months before the diagnosis was made; they were already present and persisted when the patient converted to malignancy which was similar as what we have found [1,2]. We have indicated the sample size in the limitations of this study.

  1. Imai, Haruhiko, et al. "Increasing titers and changing specificities of antinuclear antibodies in patients with chronic liver disease who develop hepatocellular carcinoma." Cancer 71.1 (1993): 26-35.
  2. Zhang, Jian-Ying, and Eng M. Tan. "Autoantibodies to tumor-associated antigens as diagnostic biomarkers in hepatocellular carcinoma and other solid tumors." Expert review of molecular diagnostics 10.3 (2010): 321-328

    We tried our best to improve and revise the manuscript. These changes will not influence the content and framework of the paper. We did not list the changes in our response document, but they are marked in the revised paper. We appreciate the editors/reviewers’ comments and suggestions earnestly and hope these revisions and corrections can help our manuscript to be accepted and published. We look forward to hearing from you regarding our submission. We will be happy to respond to any further questions and comments that you may have. Once again, thank you very much for your comments and suggestions.v

Reviewer 2 Report (Previous Reviewer 2)

The findings appear to be interesting and technically well performed. Specific points that the authors need to address are as follows:

1. The number of patients whose sera was used for testing should be increased.

2. The limitations associated with the detection of autoantibodies to PAX5, PTCH1, and GNA1 should be indicated.

3. Methods in addition to enzyme-linked immunosorbent assay (ELISA) should be used to analyze autoantibodies to PAX5, PTCH1, and GNA11 in sera of patients.

4. The sensitivity appears to be low and should be impoved. 

5. The authors should provide their own justification and relevance of the study. This will help the readers to understand the importance of the paper.

Author Response

Dear Editors and Reviewers:

Thank you for your comments regarding our manuscript titled “Autoantibodies to PAX5, PTCH1, and GNA11 as serological biomarkers in the detection of hepatocellular carcinoma in Hispanic Americans”. We have read your comments carefully and made significant revisions accordingly. We have also added validation data to this manuscript. Our additional validation cohort was used to confirm the performance of the three autoantibodies in the detection of hepatocellular carcinoma in Hispanic patients. In addition, more contents were added to the manuscript. Revised portions are marked in red in the manuscript and responses to the reviewers’ comments are as follows:

Dear Editors and Reviewers:

Thank you for your comments regarding our manuscript titled “Autoantibodies to PAX5, PTCH1, and GNA11 as serological biomarkers in the detection of hepatocellular carcinoma in Hispanic Americans”. We have read your comments carefully and made significant revisions accordingly. We have also added validation data to this manuscript. Our additional validation cohort was used to confirm the performance of the three autoantibodies in the detection of hepatocellular carcinoma in Hispanic patients. In addition, more contents were added to the manuscript. Revised portions are marked in red in the manuscript and responses to the reviewers’ comments are as follows:

Response to Reviewer 2

Comments to the Author

The findings appear to be interesting and technically well performed. Specific points that the authors need to address are as follows:

  1. The number of patients whose sera was used for testing should be increased.

Response: We thank the reviewer for this comment. We have added an independent cohort including 53 HCC, 35 LC, 35 CH, and 43 NC to our study. Unfortunately, there are no more Hispanic sera available in our serum bank, so additional non-Hispanic sera were used. Interestingly, one of the three autoantibodies has no significant difference among the HCC patients at different stages and normal controls in non-Hispanics, which may indicate that this autoantibody might be specific to Hispanics.

  1. The limitations associated with the detection of autoantibodies to PAX5, PTCH1, and GNA1 should be indicated.

Response: We thank the reviewer for this comment. The limitations of this study have been added to the manuscript.

  1. Methods in addition to enzyme-linked immunosorbent assay (ELISA) should be used to analyze autoantibodies to PAX5, PTCH1, and GNA11 in sera of patients.

Response: We thank the reviewer for this comment. ELISA is a common method for detecting autoantibodies, which is designed to rapidly handle a large number of samples in parallel with high sensitivity and specificity. In this study, we mainly used this method for autoantibody determination with strict quality control. Eight diluted sera were added to every plate for normalization among different plates, if the coefficient of variation (CV) value among different plates is more than 15.0%, we would re-do the experiment. Our ELISA was done according to regular procedures. Due to the limited volume of serum samples, we were unable to analyze these autoantibodies with other methods. We will consider and implement your suggestions in our future study. Thanks!

  1. The sensitivity appears to be low and should be improved.

Response: We thank the reviewer for this comment. The sensitivity of a single autoantibody is usually lower than the combination of multiple autoantibodies as a panel. Therefore, we combined the three autoantibodies together as a parallel test to enhance the sensitivity of HCC detection to 68.0% which is as high as what the AFP reached. We believe that the sensitivity of HCC detection would be further improved with the combination of the three-autoantibody panel and AFP.

  1. The authors should provide their own justification and relevance of the study. This will help the readers to understand the importance of the paper.

Response: We thank the reviewer for this comment. Although biomarkers related to cancer are well studied, autoantibodies associated to cancer and the three autoantibodies were reported in a previous study; their existence and specificity among different ethnic populations are rarely reported. Moreover, their detection in patients who are transitioning from high-risk to HCC has not been seen yet This is the first study to evaluate autoantibodies to PAX5, PTCH1, and GNA11 in the Hispanic populations including different groups of patients corresponding to various stages of HCC development. In this study, we found that the three autoantibodies have already been presented in pre-malignant lesions prior to clinical diagnosis, and they could identify liver cirrhosis patients at high risk of HCC from normal controls. We also analyzed the three autoantibodies in an independent non-Hispanic cohort to evaluate their specificity and it showed that autoantibody to PTCH1 may be specific to Hispanic populations. We have added information regarding our justification and the relevance of the study to the introduction and discussion of the manuscript. We hope that our study can provide valuable evidence for future research.

We tried our best to improve and revise the manuscript. These changes will not influence the content and framework of the paper. We did not list the changes in our response document, but they are marked in the revised paper. We appreciate the editors/reviewers’ comments and suggestions earnestly and hope these revisions and corrections can help our manuscript to be accepted and published. We look forward to hearing from you regarding our submission. We will be happy to respond to any further questions and comments that you may have. Once again, thank you very much for your comments and suggestions.

Round 2

Reviewer 1 Report (New Reviewer)

The authors edited all inquiries and the manuscript now is appropriate for publication

This manuscript is a resubmission of an earlier submission. The following is a list of the peer review reports and author responses from that submission.

Round 1

Reviewer 1 Report

LINE 102: signif-icantly

LINE 105: com-bine

LINE 106: signifi-cantly

LINE 225-226: "it was further confirmed that the levels of these three autoantibodies have risen in nine to three months before being diagnosed with HCC". With 5 out of 8 patients, perhaps a sentence like "in our small sample" would give a much  realistic perception of your findings.

Reviewer 2 Report

The findings appear to be interesting and technically well performed. Specific points that the authors need to address are as follows:

1. The number of patients whose sera was used for testing should be increased.

2. The limitations associated with the detection of autoantibodies to PAX5, PTCH1, and GNA1 should be indicated.

3. Methods in addition to enzyme-linked immunosorbent assay (ELISA) should be used to analyze autoantibodies to PAX5, PTCH1, and GNA11 in sera of patients.

4. The sensitivity appears to be low and should be improved. 

5. The authors should provide their own justification and relevance of the study. This will help the readers to understand the importance of the paper.